# Analysis of Expression of Programmed Cell Death Ligand 1 (PD-L1) and *BRAF^V600E^* Mutation in Thyroid Cancer

**DOI:** 10.3390/cancers15133449

**Published:** 2023-06-30

**Authors:** Mizuki Sekino, Manabu Iwadate, Yukie Yamaya, Yoshiko Matsumoto, Satoshi Suzuki, Hiroshi Mizunuma, Keiichi Nakano, Izumi Nakamura, Shinichi Suzuki

**Affiliations:** 1Department of Thyroid and Endocrinology, Fukushima Medical University, 1 Hikariga-oka, Fukushima City 960-1295, Japan; semizuki@fmu.ac.jp (M.S.); iwadate@fmu.ac.jp (M.I.); yukie-y@fmu.ac.jp (Y.Y.); a0001568@fmu.ac.jp (Y.M.); satop@fmu.ac.jp (S.S.); mizunuma@fmu.ac.jp (H.M.); kei-n@fmu.ac.jp (K.N.); izumin@fmu.ac.jp (I.N.); 2Department of Thyroid Treatment, Fukushima Medical University, 1 Hikariga-oka, Fukushima City 960-1295, Japan

**Keywords:** programmed cell death 1 ligand 1 (PD-L1), *BRAF^V600E^* mutation, thyroid cancer, CD8-positive T-lymphocytes, immunohistchemistry

## Abstract

**Simple Summary:**

Immune checkpoint inhibitors are expected to be used in clinical practice to treat thyroid cancer. Programmed cell death ligand 1 (PD-L1) is the ligand expressed on the surface of tumor cells. Recent studies have reported that PD-L1 overexpression can impede T cell activation and result in tumor growth. In thyroid cancer, it has been suggested that PD-L1 overexpression is associated with some clinicopathological factors and prognosis. However, the characteristics of the tumor microenvironment of thyroid cancer or expression of PD-L1 have not yet been clarified; the effectiveness of anti-PD-L1 antibody to thyroid cancer is unclear. We found that PD-L1 expression is associated with *BRAF^V600E^* mutation, CD8+ expression and low T cell activation, suggesting a mechanism of tumor evasion from the immune surveillance system in thyroid cancer. This new finding suggests that immune checkpoint inhibitors can be also expected to be effective in thyroid cancer.

**Abstract:**

In thyroid cancer, it has been suggested that PD-L1 overexpression is associated with some clinicopathological factors and prognosis. The aim of this study is to characterize the expression of PD-L1, the presence of the *BRAF^V600E^* mutation, as well as cellular and humoral immunity in thyroid cancer, and to investigate the factors that predict the effectiveness of anti-PD-L1 antibody therapy. Blood samples were collected from 33 patients who were newly diagnosed with thyroid cancer after surgery or biopsy. PD-L1 expression, *BRAF^V600E^* mutation, and CD8+ expression were examined by immunohistological staining using clinical thyroid cancer specimens. With a PD-L1 staining cut-off value of 1%, 13 (39.4%) patients were classified as PD-L1 positive. Stimulation Index (SI) is an indicator of T cell activation. PD-L1 expression was significantly correlated with low SI level (*p* = 0.046). Moreover, *BRAF^V600E^* mutation was detected in 24 of the 33 (72.7%) patients, and was significantly associated with PD-L1 expression (*p* = 0.047). In addition, enhanced CD8+ expression was significantly associated with PD-L1 expression (*p* = 0.003). Multivariate analyses confirmed that high CRP levels (*p* = 0.039) were independently and significantly associated with poor progression-free survival. These findings suggest that elevated PD-L1 status can be a prognostic indicator for survival in patients with thyroid cancer when comprehensively assessed using the expression of CD8+, the presence of *BRAF^V600E^* mutation and the patient’s immune status.

## 1. Introduction

Differentiated thyroid carcinomas (DTCs) include papillary thyroid carcinomas (PTCs) and follicular thyroid carcinomas (FTCs), and account for about 95% or more of thyroid carcinomas. The majority of DTCs have a good prognosis after treatment, but some DTCs have a poor prognosis, with repeated recurrence and metastasis. On the other hand, anaplastic thyroid cancer (ATC), one of the most lethal human solid cancers, has a median survival from diagnosis of about 6 months, and a one-year survival rate of less than 30%. ATC has been reported to be associated with the immune system and inflammatory conditions [1]. Blood tests often show leukocytosis, increased red sedimentation, and elevated c-reactive protein (CRP) levels. Recently, CRP and neutrophil–lymphocyte ratio (NLR) have been reported to be associated with cancer grade and poorer prognosis in not only ATC, but also DTC [2,3].

In addition, immune checkpoint inhibitors are also expected to be used in clinical practice to treat thyroid cancer. A phase Ib KEYNOTE-028 trial conducted to evaluate the safety and efficacy of pembrolizumab, an anti-PD-L1 antibody, evaluated patients with advanced differentiated thyroid cancer expressing programmed cell death ligand 1 (PD-L1) [4]. Several studies have shown the usefulness of PD-L1 expression in tumor cells [5,6], the amount of gene mutations in tumor [7], and lymphocytes that migrate to the cancer microenvironment [8] as biomarkers for selecting patients who are sensitive to cancer immunotherapy. However, biomarkers that can determine in advance the sensitivity or insensitivity of immune checkpoint inhibitors to thyroid cancer are lacking.

CD8 is a cell surface marker expressed on T cells. CD8+ T cells, which express CD8 co-receptor on their surface, are also known as cytotoxic T cells. CD8+ T cells recognize specific antigens presented by the MHC class I molecules on antigen-presenting cells, and upon activation, they target and kill infected cells or tumor cells. On the other hand, PDL1 is a protein that normally appears on the surface of antigen-presenting cells. The ligand for PDL1, PD1, is expressed on the surface of T cells. When PDL1 binds to PD1, T cell production of cytokines is reduced, and a signal is transmitted that suppresses T cell activity. PDL1 is also strongly expressed in tumor cells. When this PDL1 binds to PD1 on the surface of activated cytotoxic T cells, T cell activity is suppressed. Inactivated T cells then remain in the tumor microenvironment without migrating. Indeed, PD-L1 expression levels have been shown to be correlated with T cell infiltration [9], and such PD1/PDL1-mediated mechanisms represent tumor cell resistance to tumor immunity [10].

In recent years, it has been found that PDL1 is overexpressed in cancers such as non-small cell lung cancer (NSCLC), malignant melanoma, and renal cell carcinoma. Overexpression of PDL1 has been reported to be associated with resistance to anti-cancer therapy and poor prognosis [11,12,13]. Inhibitors that block immune checkpoints have been developed with the aim of weakening such resistance to tumor immunity and maintaining the body’s natural immune response to tumors. PD-L1 overexpression is found in approximately 46.4% of patients with thyroid cancer, and has been suggested to be a predictor of poor prognosis and recurrence, like in other cancers [14,15]. However, there are few studies on the prognostic significance of PD-L1 expression in thyroid cancer [16,17,18,19,20,21], and said significance remains controversial because of differences of the antibodies, scoring methods, and targeted histological types among said studies. The functions of cytotoxic T cells and helper T cells are regulated by many checkpoint molecules other than PD-L1, and the tumor microenvironment differs depending on the carcinoma. However, the characteristics of the tumor microenvironment of thyroid cancer and its relationship with PD-L1 have not been clarified. Therefore, it is unclear whether anti-PD-L1 antibody has the same therapeutic effect as malignant melanoma and non-small cell lung cancer, because there may be an immune escape mechanism unique to thyroid cancer.

Surgery is the first choice of thyroid cancer, while in high-risk cases, postoperative adjuvant therapies such as radioactive iodine therapy and thyroid hormone therapy are also employed. In recent years, there have been advancements in the development of targeted therapies for difficult-to-cure thyroid cancers, including radioactive iodine-refractory thyroid cancer, as a novel treatment approach [22]. The status of genetic mutations in thyroid cancer has become clearer, and it has been reported that certain genetic mutations influence the progression of thyroid cancer [23]. *BRAF^V600E^* mutation, *RAS* mutation, *RET*/*PTC* gene rearrangement, and *TERT* and *p53* mutations are closely related to the characteristics and malignancy of thyroid cancer [24]. Some studies have reported that the *BRAF^V600E^* mutation is associated with indoleamine 2, 3-dioxygenase (IDO), human leukocyte antigen-G (HLA-G), PD-L1 expression, and a low CD8+ effector T cell to FoxP3+ suppressor T cell ratio [20,25,26]. Thyroid cancer encompasses a wide range of tumor types, with not only genetic mutations but also diverse molecular, histological, and clinical characteristics. Even patients with the same thyroid cancer type can select effective treatments depending on the type of genetic mutation and molecular characteristics. Therefore, to make risk-adapted and personalized treatments more available than ever before, it is necessary to identify as many powerful biomarkers as possible.

The aim of this study is to characterize the expression of PD-L1, the presence of the *BRAF^V600E^* mutation, and cellular and humoral immunity in thyroid cancer. Furthermore, the aim is to investigate whether specific factors can serve as indicators to enable the selection of anti-PD-L1 antibody therapy in thyroid cancer.

## 2. Materials and Methods

### 2.1. Patients

We used the data of blood samples and archival samples from all 33 patients with thyroid cancer who were newly diagnosed as having thyroid cancer and had undergone surgery or biopsy at Fukushima Medical University between January 2011 and January 2013 [27,28]. The age of patients at the time of surgery or biopsy was 57.6 ± 13.9 years (median 58 years, range 24–90). Thirty-three patients had histologically confirmed thyroid cancer, including twenty-four patients with PTC, five with ATC, two with medullary thyroid cancer (MTC) and two with FTC. All five patients with ATC exhibited undifferentiated transformation of PTC. We assessed the histological subtypes/variants of PTC using the 7th edition of the *Japanese General Rules for the Description of Thyroid Cancer*, which is based on the seventh classification of the Union for International Cancer Control (UICC). Of the 33 patients, 10 had stage I disease, 1 had stage II, 4 had stage III, and 18 had stage IV. For the four patients who did not undergo surgery, we determined their pT, pN, and stagings based on results from biopsy and ultrasonography.

This study was approved by the ethical committee of Fukushima Medical University (approval no. 29195), which is guided by local policy, national law, and the World Medical Association Declaration of Helsinki.

### 2.2. Immunohistochemistry

Four μm sections of formalin-fixed paraffin-embedded tissues of thyroid cancer were stained with Hematoxylin and Eosin stain, and underwent immunohistochemistry (IHC) analysis. IHC was carried out using the following primary antibodies: an anti PD-L1 (clone E1L3N; 1/200; Cell Signaling Technology, Inc., (CST), Danvers, MA, USA), anti *BRAF^V600E^* (clone VE1; 1/500; Spring Bioscience, Pleasanton, CA, USA), and anti CD8 (clone 4B11; ready to use; Leica BIOSYSTEMS Ltd., Newcastle, UK). We conducted IHC using a Bond Polymer Refine Detection kit (Leica Biosystems Ltd., Newcastle, UK) on an automated staining platform (BOND-III, Leica Biosystems Ltd., Newcastle, UK), following the manufacturer’s instructions [29].

### 2.3. Scoring of Immune Markers

PD-L1 and CD8 were analyzed using image analysis software Patholoscope at 400 magnification (Figure 1 and Figure 2). Since there are no established means to determine the PD-L1 expression rate or a cutoff value for positive expression using the PD-L1 antibody clone E1L3N, in this study, the rate of PD-L1 expression was determined by calculating the Tumor Proportion Score (TPS) based on a previous study [30]. First, the expression of PD-L1 was determined by membranous staining of tumor cells. Then, the rate of PD-L1 expression was determined using the Tumor Proportion Score (TPS), which is the proportion of tumor cells expressing PD-L1 among the total number of tumor cells. It was calculated as the number of tumor cells expressing PD-L1 divided by the total number of tumor cells, multiplied by 100%. A high value indicates strong expression of PD-L1, while a low value indicates weak or no expression of PD-L1. Moreover, based on the phase Ib KEYNOTE-028 trial, the cut-off value was determined to be 1% [4]. Patients were categorized into one of two groups according to a PD-L1 cut-off level of 1%; hence, <1% was designated as PD-L1-negative and ≥1% was considered PD-L1-positive. We determined CD8-positive if the entire cytoplasm was stained. Three non-overlapping fields on the HE-stained slides were selected, and the number of positive lymphocytes in all three fields was averaged [31,32]. The median value was used to separate the patient cohort into two groups; those with negative CD8 expression and those with positive CD8 expression.

### 2.4. Flow Cytometry

We analyzed peripheral blood mononuclear cells (PBMC) using flow cytometry to identify MDSCs, which were defined as CD11b + CD14 − CD33+ cells. Blood samples were labeled with fluorescein isothiocyanate (FITC), phycoerythrin (PE), and phycoerythrin-cyanin 5.1 (PC5), using antibodies directed against CD14 (Abcam, Cambridge, UK), CD11b (Beckman Coulter, Marseille, France), and CD33 (Beckman Coulter), respectively. The antibodies were diluted in phosphate-buffered saline (PBS) to 10 and 50 µg/mL, and cells were incubated with the antibodies at 4 °C for 20 min, then washed with PBS. Data acquisition and analysis were performed using a FACSAria II flow cytometer (BD Biosciences, Mountain View, CA, USA) and analyzed using FlowJo software (TreeStar Inc., Ashland, OR, USA) [27].

### 2.5. Cytokine Production by PBMCs

Around 106 PBMCs were cultured in 1 mL of RPMI-1640 medium with 10% heat-inactivated fetal calf serum (Gibco BRL, St. Louis, MO, USA) and 100 µg/mL phytohemagglutinin (PHA; Sigma, Rockville, MD, USA) at 5% CO_2_ and 37 °C for 24 h. Afterwards, the supernatants were collected and frozen at −80 °C until further use. To measure the concentrations of vascular endothelial growth factor (VEGF), interleukin (IL)-10, IL-12, IL-17, and IFN-γ, enzyme-linked immunosorbent assay (ELISA) kits from R&D Systems (Minneapolis, MN, USA) were utilized. Each sample was only used once after being thawed [27].

### 2.6. Lymphocyte Proliferation Assay

To assess lymphocyte proliferation, PBMCs were suspended in RPMI-1640 medium (Wako Pure Chemical Industries, Osaka, Japan) with 10% fetal calf serum (Sigma-Aldrich, St. Louis, MO, USA). The PBMC culture wells were then incubated with 10 µg/mL PHA at 37 °C in a 5% CO_2_ atmosphere for 80 h to induce PHA mitogenesis. During the last 8 h of incubation, 3H-thymidine (Japan Radioisotope Association, Tokyo, Japan) was added to the wells. The harvested cells were analyzed for 3H-thymidine incorporation using a liquid scintillation counter (PerkinElmer Inc., Waltham, MA, USA), and the results were expressed as counts per minute (CPM). To obtain the stimulation index (SI) for PHA-induced lymphocyte blastogenesis, the total CPM was divided by the control CPM. PBMCs that were not exposed to PHA were used as controls [27].

### 2.7. Markers of Inflammation and Thyroid Status

CRP, WBC, neutrophil lymphocyte rate (NLR), and lymphocyte to monocyte ratio (LMR) were used as indicators of inflammation. Thyroid status was evaluated with thyroid hormone (FT4, FT3, ATG, ATPO, HTG).

### 2.8. Statistical Analysis

The data presented in this study are reported as the mean values along with their corresponding standard deviations. The statistical significance of differences between groups was evaluated using Student’s *t*-test, and *p* < 0.05 was considered to indicate statistical significance. For cases where insufficient blood samples were collected from patients, certain measurements were not feasible. Fisher’s exact test, chi-square test, or Mann–Whitney U test was used as appropriate for statistical analysis. For patients who were alive and had not experienced relapse, the date of their last follow-up visit was used as the censoring date. Progression-free survival (PFS) was defined as the time interval between the date of surgery or biopsy and the date of disease progression, metastases, or death, and was calculated using the Kaplan–Meier method. The significance of differences in PFS was assessed using the log-rank test. Additionally, a Cox proportional hazard model was utilized for multivariate regression analysis. All statistical analyses were conducted using SPSS Statistics for Windows v26.0 (SPSS, Inc., Chicago, IL, USA).

## 3. Results

### 3.1. Correlation between PD-L1 Expression in Cancer Tissues and Clinicopathologic Factors

Characteristics of all patients (*n* = 33) are shown in Table 1. With a PD-L1 staining cut-off value of 1%, 20 (60.6%) patients were classified as negative (<1%), and the remaining 13 (39.4%) patients were classified as positive (≥1%). Stimulation Index (SI) is an indicator of T cell activation, and PD-L1-positive was positively associated with low SI levels (*p* = 0.046). Although no significant difference was observed, in the PD-L1-positive patients, the production of VEGF and IL-10 and MDSC that were immunosuppressive molecules were higher, and IL-12, IL-17, and INF-γ were lower, compared to the PD-L1-negative patients. The expression of PD-L1 was not related with tumor size, invasion, pTNM factors and UICC stagings, or any pathological features (*p* > 0.05) (Table 1).

### 3.2. Correlation between PD-L1 Expression and BRAF^V600E^ Mutation

In this study, 24 (72.7%) patients were found to be positive for the *BRAF^V600E^* mutation, and nine (27.3%) were found to be positive for *BRAF* wild type. In the patients with *BRAF^V600E^* mutation, 22 had PTC and two had ATC, and none were mutated with FTC or MTC (*p* = 0.001). The *BRAF^V600E^* mutation was significantly associated with increasing expression of PD-L1. Twelve (50%) of twenty-four patients with *BRAF^V600E^* mutation were PD-L1 positive, compared to one (11.1%) of nine patients with *BRAF* wild type (*p* = 0.047) (Table 1).

### 3.3. Correlation between PD-L1 Expression and CD8+ Expression

When the median value of the number of positive T cells with CD8+ was classified as a cut-off, 16 (48.5%) patients out of the total 33 were negative and 17 (51.5%) patients were positive. CD8+ expression was significantly correlated with increased PD-L1 expression. Of the 17 patients with CD8+-positive, 11 (64.7%) were PD-L1 positive, compared to six (35.3%) CD8+ negative patients (*p* = 0.003) (Table 1).

### 3.4. Correlation between PD-L1 Expression in Thyroid Cancer and Patient Survival

The correlation between PFS and characteristics was examined using Cox proportional hazard model. The median follow-up period for cases was 55 months (range 0–92), during which 15 patients (45.5%) experienced tumor persistence, recurrence, or death. In the univariate analysis, several factors were independent predictors for poor PFS, including high CRP (HR = 5.08 [CI 1.26–64.76], *p* = 0.002), high WBC (HR = 30.93 [CI 2.81–340.82], *p* = 0.005), high VEGF (HR = 9.02 [CI 1.559–6.383], *p* = 0.029), primary tumor size ≥20 mm (HR = 7.79 [CI 1.74–34.78], *p* = 0.007), ATC (HR = 4.97 [CI 1.53–16.18], *p* = 0.008), extrathyroidal extension (HR = 9.3 [CI 2.28–37.9], *p* = 0.002), and distant metastasis (HR = 5.73 [CI 1.98–16.58], *p* = 0.001) (Table 2). On the other hand, patients with *BRAF^V600E^* mutation had a significantly favorable PFS (HR = 0.27 [CI 0.10–0.76], *p* = 0.013). There were no significant differences between elevated PD-L1 expression and PFS (HR = 1.04 [CI 0.37–2.91], *p* = 0.947) (Table 2).

In the multivariate analysis, all significant factors identified in the univariate analysis, including PD-L1 expression, were included. The results showed that *BRAF* wild type (*p* = 0.022) and high CRP (*p* = 0.039) were independent predictors for poor PFS (Table 3).

The Kaplan–Meier curve of *BRAF^V600E^* mutation showed a significant association with favorable PFS (*p* = 0.007), while the curve of PD-L1 expression and CD8+ expression showed no significant differences. However, these results suggested a potential trend towards poorer prognosis in CD8+ negative patients (Figure 3). Furthermore, we separated the 33 patients into four groups based on PD-L1 and CD8 expression (positive or negative) and *BRAF^V600E^* mutation or wild type, and subjected to sub-analyses. It was suggested that patients who were PD-L1-positive/CD8+-negative may have a tendency towards poor prognosis (Appendix A).

## 4. Discussion

We conducted IHC for PD-L1 expression in tumor cells and CD8+ infiltration within the tumor to investigate tumor immune status, and investigated the presence of *BRAF^V600E^* mutation which was the most representative genetic mutation among thyroid cancer, using surgical specimens from initial surgeries of 33 patients with thyroid cancer in our department. Blood samples were also collected from all 33 patients to measure immune suppressive cytokines, immune suppressive cells, markers of inflammation, and thyroid hormone levels. Furthermore, we evaluated the correlation between these factors and PD-L1 expression to investigate whether specific factors can serve as indicators to enable the selection of anti-PD-L1 antibody therapy in thyroid cancer. In our current study, 39.4% of the subjects were PDL1-positive.

In this study, there was no correlation between PD-L1 expression and clinicopathological factors such as tumor size, pTNM classification, or late-stage, which are known to be prognostic, and risk factors for recurrence of thyroid cancer as reported by Ahn S, et al. [25]. However, there is still a possibility of variation in the relationship between PD-L1 expression and telomerase reverse transcriptase (*TERT*) promoter mutation and *RET/PTC*, which were not tested in this study. Furthermore, the sample size in our study was limited. As each thyroid cancer has distinct clinical characteristics, it is necessary to increase the sample size and analyze the relationship between clinical–pathological variables and PD-L1 expression for each thyroid cancer subtype. Moreover, PD-L1 expression was not associated with prognosis. Previous studies have shown inconsistent findings regarding the association between PD-L1 expression and prognosis [1,14,16,25]. One reason for the discrepancies is the variability in the cutoff values used to classify PD-L1 expression as high or low. The optimal cutoff value for PD-L1 expression has not yet been validated for thyroid cancer [4]. The cutoff values currently used include 0%, 1%, 5%, 10%, 25%, or 30% and higher. In this study, we defined the cutoff value as 1% based on previous research [30]. Additionally, differences in the antibodies, experimental protocols, and scoring methods used to establish these cutoff values may also contribute to the discordance in results. We have been using the PD-L1 antibody clone E1L3N, having assessed its staining pattern for some time [33]. Because it is positioned in the Laboratory Developed Test (LDT), the criteria for stainability and the cutoff value have not been determined. Comparing the stainability of clone E1L3N with clone 28–8 and clone SP142, it was reported that they showed high degree of coincidence (Kappa coefficient = 0.69) [29]. Therefore, we trust that there was no problem in using clone E1L3N in this study. In order to apply clone E1L3N clinically, it is necessary to manage the precision of immunohistochemistry and standardize the scoring.

The present study also showed that patients who were PD-L1-positive had significantly higher levels of CD8+-positive expression. CD8+ T cells have been identified as significant factors for antitumor immunity. In fact, IHC images appear to show co-localization of PD-L1 staining and CD8 staining. In tumors rich in T cells, it has been shown that T cell infiltration into the tumor is activated and INF-γ is increased by the recognition of cancer cells [9]. Thus, we showed that, also in thyroid cancer, the expression of PD-L1 is upregulated to suppress the immune system from attacking cancer cells when T cells increase in the tumor microenvironment. Chronic exposure to antigens and immunosuppressive cells such as MDSCs can cause T cell dysfunction. The survival analysis in this study suggested that the PDL1-positive/CD8-negative group had a poor prognosis, indicating that the tumor immune response was suppressed. In such cases, immune checkpoint inhibitors can be expected to reactivate the immune response.

Stimulation Index (SI) is an indicator of T cell activation, and PD-L1-positivity was positively associated with low SI levels (*p* = 0.046). In the multivariate analysis, the results revealed that high CRP (*p* = 0.039) was an independent and significant predictive factor for poor PFS. CRP has been reported to be associated with cancer grade and poorer prognosis in not only ATC but also DTC [2,3]. Persistent inflammation and repeated exposure to tumor antigens in the tumor microenvironment gradually increase PD-1 activity and exhaust T cells. Due to the low activity level of T cells, there may be a possibility that the immune response against tumors is insufficient. Immunotherapy, such as immune checkpoint inhibitors, antibody therapy, and immunomodulatory therapy, can be effective in patients with low T cell activation levels by promoting T cell activation and inducing an appropriate immune response against tumors.

Moreover, a higher frequency of CD8+ T cells was found to be associated significantly with favorable survival in multiple tumors, including differentiated thyroid cancer [26,34,35]. This study also showed that CD8+ expression was significantly associated with prognosis. Further analysis revealed that the group with PD-L1-positive/CD8+-negative tended to have poorer prognosis. Combining the evaluation of CD8+ expression and PD-L1 expression may serve as a predictive biomarker for prognosis [17,18], and more research is needed.

The *BRAF^V600E^* mutation was significantly associated with increasing expression of PD-L1. It is suggested that the *BRAF^V600E^* mutation in thyroid cancer promotes immunosuppression and contributes to cancer immune evasion. On the other hand, it has been reported that there is no difference in prognosis between *BRAF^V600E^* mutation and *BRAF* wild type in Japan [36]. However, contrary to these findings, the present study showed that *BRAF^V600E^* mutation was independent and a significant predictive factor of favorable PFS. For one reason, the *BRAF* wild-type population in this study could include patients with aggressive fusion genes, such as *RET/PTC* and *NTRK*. Furthermore, it was possible that the difference in concept of noninvasive follicular thyroid neoplasm with papillary-like nuclear features (NIFTP) between Japan and Western countries was reflected [37]. The major genetic mutation in NIFTP is the *RAS* mutation (29.6–59%) [38]. Therefore, it was suggested that the prognosis of *BRAF* wild type including NIFTP might be better than that of *BRAF^V600E^* mutation in Western countries. The relationship between *BRAF^V600E^* mutation and prognosis requires further investigation, and analysis is ongoing.

Our study had certain limitations, including its retrospective nature, relatively short median follow-up, and small sample size. The sample size was small, because the subjects of this study were further selected from those who participated in the previous study [27]. Prospective studies with larger sample sizes are needed to evaluate the correlation between PD-L1 expression, clinical response, and survival, and to establish the predictive value of PD-L1 expression, CD8+ TILs, and *BRAF^V600E^* mutation in thyroid cancer. Moreover, PD-L1 expression was analyzed at one selected hotspot rather than the entire tumor section, so it was considered that the heterogeneity in the tumor was not completely covered. Tumor genotype is associated with a particular subtype of thyroid cancer [39]. One of the reasons is that the amount of cancer-associated transcription factors varies from cell to cell. Norman G et al. revealed that the expression patterns of the transcription factors SP1 and TCF7L2, which were involved in gene expression, differed depending on types of FTC invasion. Moreover, in IHC, the factors tended to enhance expression along the tumor infiltration front relative to the central tumor [40]. Therefore, it may not be possible to make a precise diagnosis or predict prognosis by analyzing only one part of the heterogeneous tumor tissue, and we need to consider the method for individually analyzing cells composing tumors.

While there are many challenges ahead, it is a novel finding that comprehensive evaluation of factors such as PD-L1 expression, CD8 expression, *BRAF^V600E^* mutation, and the patient’s immune status can serve as prognostic indicators. It suggested the presence of mechanisms in which the immunological composition of the tumor microenvironment and inflammatory conditions influence the progression of thyroid cancer through PD-L1. Furthermore, by combining the evaluation of PD-L1 expression, CD8 expression, and *BRAF^V600E^* gene mutation status, it is possible to achieve more individualized patient risk classification. Additionally, the study presented results of PD-L1 expression in not only papillary thyroid carcinoma but also undifferentiated carcinoma, follicular carcinoma, and medullary carcinoma. In follicular carcinoma and medullary carcinoma, all patients showed negative PD-L1 expression. Although the sample size was small and no significant differences were observed, these data provide valuable insights as there are no prior studies reporting data on PD-L1 expression in thyroid cancers other than papillary and undifferentiated carcinoma.

As a prospect for defining risk classes of patients with thyroid cancer, we expect the development of predictive models that combine clinical, pathological, and genetic data. One useful way to make this feasible would be liquid biopsy. Liquid biopsy is a technique that involves collecting bodily fluids, such as blood, from patients and analyzing the tumor cells or substances derived from tumor cells present in the fluid. The advantage is that it can serve as an alternative when tumor tissue is not obtainable. Furthermore, it has the potential to detect disease progression and treatment resistance before clinical symptoms arise or before they can be identified through diagnostic means. Compared to traditional fine-needle aspiration cytology, liquid biopsy is less invasive, a short procedure, and can provide more comprehensive molecular information [41]. Circulating tumor DNA which encompasses genetic mutations and abnormal methylation status, the expression levels of circulating tumor RNA, gene expression of circulating tumor cells and epithelial cells, and the expression levels of exosome-derived RNA in the extracellular vesicles, are analyzed to extract molecular information from patients. These markers have been utilized as indicators for diagnosis, prognosis, and treatment response in major types of thyroid cancer such as DTC, ATC, and MTC, demonstrating their utility [41]. Although there are no reports specifically examining the correlation between peripheral blood data and local data in thyroid cancer. In Japan, liquid biopsy is currently approved and covered by insurance for non-small cell lung cancer, colorectal cancer, and solid tumors. Despite the challenges, we believe liquid biopsy can serve as a valuable resource to support clinical decision-making in patients with thyroid cancer by aiding in the definition of diagnosis, prognosis prediction, assessment of treatment response, and disease management.

## 5. Conclusions

We conducted IHC to examine the expression of PD-L1, CD8, and the presence of *BRAF^V600E^* mutation in thyroid cancer. The results showed that 39.4% of all patients exhibited positive expression of PD-L1. It was found that the presence of *BRAF^V600E^* mutation, positive expression of CD8, and decrease in T cell activity were associated with PD-L1 positivity. Although no clear associations were shown, there was a tendency for better prognosis when there was a higher infiltration of CD8+ T cells within the tumor, while it was not demonstrated that high expression of PD-L1 in tumor cells was an independent poor prognostic factor. However, comprehensive evaluation of PD-L1 expression, CD8 expression, *BRAF^V600E^* mutation, and the patient’s immune status suggested the potential of serving as predictive indicators for the prognosis. In other words, the immune status of the tumor microenvironment in thyroid cancer is an important factor that influences patient prognosis. The results showed that the immunological composition of the tumor microenvironment and inflammatory conditions are influencing the progression of thyroid cancer through the involvement of PD-L1. It was suggested that immune response suppression factors such as T cell dysfunction and exhaustion, decreased production of cytokines, the presence and activity of immunosuppressive cells (MDSCs), chronic inflammatory states, and tumor-associated inflammation contribute to immune response suppression. Further research is necessary to understand the mechanisms underlying PD-L1 expression.

By comprehensively evaluating factors such as PD-L1 expression, CD8 expression, *BRAF^V600E^* mutation, and the patient’s immune status, it may be possible to stratify patients who are likely to benefit from anti-PD-L1 antibody therapy and propose effective treatment combinations. For example, even among PD-L1-positive patients, there is a trend suggesting that prognosis may vary depending on whether CD8 is positive or negative, indicating differences in T cell activity. From this observation, it can be inferred that PD-L1-positive and CD8-negative patients may have impaired tumor cell attack by T cells due to upregulation of PD-L1. Therefore, in addition to inhibiting PD-L1 with anti-PD-L1 antibody therapy, considering the selection of therapies such as CAR-T cell therapy to activate T cells, 4-1BB agonist antibody, and OX40 agonist antibody may be worthwhile.

Additionally, preoperative CRP levels, which have been identified as prognostic factors, may serve as useful markers in the development of novel therapies such as targeted molecular agents. Continued investigation is warranted to evaluate their utility in future therapeutic advancements. Further research is needed to elucidate the prognostic and therapeutic value of PD-L1 expression in thyroid cancer.

## Figures and Tables

**Figure 1 cancers-15-03449-f001:**
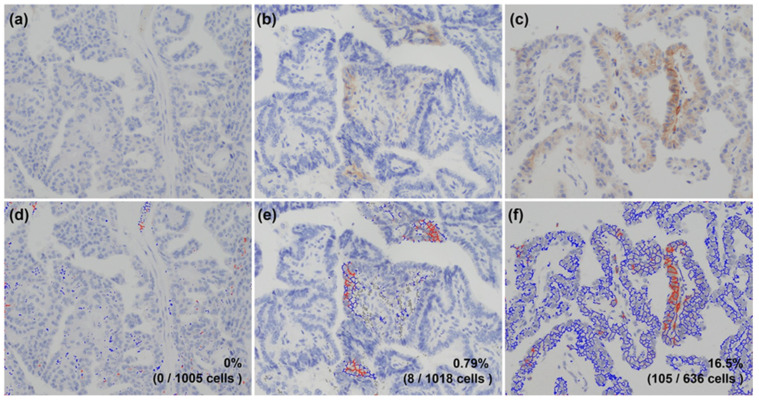
PD-L1 IHC and image analysis. PD-L1 IHC was analyzed using image analysis software Patholoscope at 400 magnification. (**a**–**c**) PD-L1 expression in papillary thyroid cancer by immunohistochemistry. (**d**–**f**) In the immunostaining image format, cells stained red indicate strong expression of PD-L1, and blue indicates weak expression of PD-L1. Both red and blue cells were counted automatically. (**a**,**d**) The percentage of PD-L1-positive staining was 0%. (**b**,**e**) The percentage of PD-L1-positive staining was 0.79%. (**c**,**f**) The percentage of PD-L1-positive staining was 16.5%. All photomicrographs were magnified ×200.

**Figure 2 cancers-15-03449-f002:**
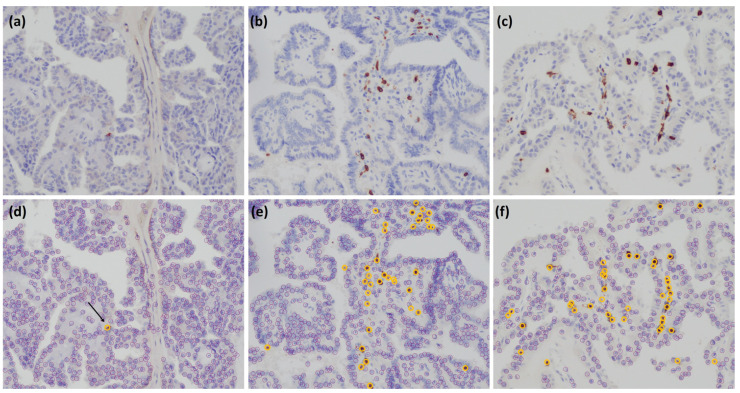
CD8+ IHC and image analysis. CD8+ IHC was analyzed using the software Patholoscope, v1.0 (MITANI Co., Tokyo, Japan) a computerized image analysis system. (**a**–**c**) The picture of the immunohistochemistry staining of CD8+ T-cell is shown. (**d**–**f**) From the immunostaining image format, the software extracted positive cells (small orange circles) and negative cells (small purple circles), and counted them automatically. As the orange circles were not clear when we pasted the images, the author re-emphasized the orange markings. (**a**,**d**) The tumor sections from the same patient as in Figure 1 (**a**,**d**) were used. (**b**,**e**) The tumor sections from the same patient as in Figure 1 (**b**,**e**) were used. (**c**,**f**) The tumor sections from the same patient as in Figure 1 (**c**,**f**) were used. All photomicrographs were magnified ×200.

**Figure 3 cancers-15-03449-f003:**
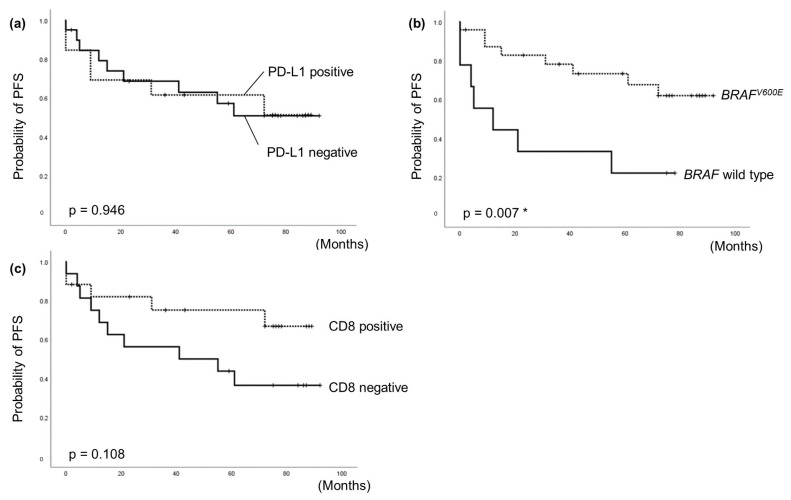
Kaplan–Meier curves for PFS; (**a**) with negative or positive PD-L1 expression, (**b**) with *BRAF^V600E^* mutation or wild type, and (**c**) with negative or positive CD8+ expression. *: statistically significant.

**Table 1 cancers-15-03449-t001:** Correlations between PD-L1 expression and clinicopathological factors in thyroid cancer.

Characteristics	PD-L1 Expression Negative (*n* = 20)	PD-L1 Expression Positive (*n* = 13)	*p*-Value
Age	53.9 ± 12.5	63.3 ± 14.5	0.071
Sex			
Male	8	2	0.132
Female	12	11	
Thyroid cancer			
PTC	14	10	0.315
ATC	2	3	
FTC	2	0	
MTC	2	0	
Tumor size	25.8 ± 20.0	34.2 ± 25.5	0.276
Ex			
Ex0	9	6	0.449
EX1	8	3	
EX2	3	4	
pT			
T1a	3	4	0.794
T1b	6	2	
T2	2	1	
T3	5	2	
T4a	1	1	
T4b	3	3	
pN			
N0	6	7	0.201
N1a	3	0	
N1b	11	6	
M			
M0	15	11	0.419
M1	5	2	
STAGE			
I	5	5	0.678
II	1	0	
III	3	1	
IV	11	7	
CRP (mg/dL)	0.19 ± 0.28	1.5 ± 5.0	0.429
WBC (10^3^/μL)	5.8 ± 3.4	7.4 ± 7.5	0.568
NLR (%)	6.9 ± 19.6	4.1 ± 4.1	0.2
LMR (%)	5.4 ± 3.4	3.9 ± 2.1	0.13
SI	694.8 ± 524.0	346.6 ± 219.0	0.046 *
IL-10 (µg/mL)	2134.9 ± 2661.8	2488.3 ± 3113.3	0.815
IL-17 (µg/mL)	834.2 ± 1785.7	480.8 ± 606.7	0.741
IL-12 (µg/mL)	116.6 ± 93.8	91.4 ± 112.3	0.256
IFN-γ (µg/mL)	3943.1 ± 2695.9	3058 ± 2321.7	0.459
VEGF (pg/mL)	369.1 ± 380.1	2246.2 ± 4617.4	0.805
MDSC (%PBMC)	1.7 ± 1.7	1.5 ± 1.3	0.626
*BRAF^V600E^*			
Wild type	8	1	0.047 *
Mutation	12	12	
CD8			
Negative	14	2	0.003 *
Positive	6	11	

Data are shown as mean ± s.d. or number of cases (percentage). *: statistically significant. PTC, papillary thyroid cancer; ATC, anaplastic thyroid cancer; FTC, follicular thyroid cancer; MTC, Medullary thyroid cancer; CRP, c-reactive protein; WBC, white blood cell; NLR, neutrophil–lymphocyte ratio; LMR, lymphocyte–monocyte ratio; SI, Stimulation Index is one of the markers of inflammation; IL, interleukin; VEGF, vascular endothelial growth factor; MDSC, myeloid-derived suppressor cells; *BRAF*, v-raf murine sarcoma viral oncogene homolog B1.

**Table 2 cancers-15-03449-t002:** Univariate analysis of clinicopathological factors associated with PFS.

Characteristics	HR	95%CI Lower	95%CI Upper	*p*-Value
Age				
<55 vs. ≥55 years	2.37	0.75	7.49	0.14
Sex				
Male vs. Female	0.57	0.2	1.01	0.290
Tumor size				
<20 vs. ≥20 (mm)	7.79	1.74	34.78	0.007 *
Thyroid cancer				
DTC vs. ATC	4.97	1.53	16.18	0.008 *
Ex				
Ex0	1	/	/	/
Ex1	3.6	0.89	14.47	0.072
Ex2	9.3	2.28	37.9	0.002 *
pT				
T1	1	/	/	/
T2	1.28	0.13	12.38	0.829
T3	3.8	0.84	17.22	0.084
T4	8.12	2.07	31.86	0.003 *
pN				
N0 vs. N1	3.45	0.97	12.25	0.056
M				
M0 vs. M1	5.73	1.98	16.58	0.001 *
Stage				
I–II vs. III–IV	3.85	0.87	17.08	0.076
CRP				
<0.15 vs. ≥0.15 (mg/dL)	5.08	1.80	14.32	0.002 *
WBC				
<8.6 vs. ≥8.6 (10^3^/μL)	30.93	2.81	340.82	0.005 *
NLR				
Low vs. High	2.08	0.69	6.23	0.191
LMR				
Low vs. High	0.58	0.20	1.69	0.321
SI				
Low vs. High	0.50	0.15	1.70	0.266
VEGF				
Low vs. High	9.02	1.26	64.76	0.029 *
IL10				
Low vs. High	0.31	0.07	1.49	0.145
IL17				
Low vs. High	0.38	0.05	2.97	0.353
IL12				
Low vs. High	0.43	0.09	2.02	0.283
IFNγ				
Low vs. High	3.16	0.88	11.29	0.077
MDSC				
Low vs. High	1.48	0.50	4.35	0.476
CD8				
Negative vs. Positive	0.43	0.15	1.26	0.122
*BRAF^V600E^*				
Wild type vs. Mutation	0.27	0.10	0.76	0.013 *
PD-L1				
Negative vs. Positive	1.04	0.37	2.91	0.947

HR: Hazard Ratio. *: statistically significant. DTC, differentiated thyroid cancer; ATC, anaplastic thyroid cancer; CRP, c-reactive protein; WBC, white blood cell; NLR, neutrophil–lymphocyte ratio; LMR, lymphocyte–monocyte ratio; SI, Stimulation Index is one of the markers of inflammation; VEGF, vascular endothelial growth factor; IL, interleukin; MDSC, myeloid-derived suppressor cells; *BRAF*, v-raf murine sarcoma viral oncogene homolog B1.

**Table 3 cancers-15-03449-t003:** Multivariate analysis of clinicopathological factors associated with PFS.

Characteristics	HR	95%CI Lower	95%CI Upper	*p*-Value
CRP				
<0.15 vs. ≥0.15 (mg/dL)	14.34	1.14	179.75	0.039 *
WBC				
<8.6 vs. ≥8.6 (10^3^/μL)	9.56	0.63	144.46	0.103
VEGF				
Low vs. high	3.83	0.31	47.41	0.295
*BRAF^V600E^*				
Wild type vs. Mutation	0.04	0.00	0.62	0.022 *
PD-L1				
Negative vs. positive	2.09	0.32	13.87	0.445

HR: Hazard Ratio. *: statistically significant. 95%CI, 95% Confidence interval; CRP, c-reactive protein; WBC, white blood cell; VEGF, vascular endothelial growth factor; *BRAF*, v-raf murine sarcoma viral oncogene homolog B1.

## Data Availability

The data presented in this study are available on request from the corresponding author.

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
