# Peer review of "Analysis of Expression of Programmed Cell Death Ligand 1 (PD-L1) and *BRAF^V600E^* Mutation in Thyroid Cancer"

_cancers, 2023, doi:10.3390/cancers15133449_

Round 1

Reviewer 1 Report

The study by Mizuki Sekino et al. is well written and interesting to read. There are, however, a few minor issues that need to be resolved before considering the study for publication:

1.       Table 2, patients characteristics:

Age <50 vs ≥50 years – why do authors use this cut-off value for patient’s age?

Tumor size <20 vs ≥20 (mm) – why this cut-off value for tumor size?

2.       Line 309 – please explain (in short) what procedure what done. Avoid generalization if possible (“to investigate immune status”)

3.       Line 323 – “One reason for the discrepancies is the variability in the cutoff values used to classify PD-L1“.

The cutoff values for the author’s study is <1% and ≥1% - can authors mention this value in this line and additionally comment and compare (in a few words) the other cutoff values?

4.       Line 346 – “Many cancer patients have some kind of barrier in the cancer-immune cycle” – please avoid this kind of generalization

5.       Line 366 – “Further analysis revealed that the group with PD-L1-positive/ CD8+ negative tended to have poorer prognosis.” – where do authors describe these findings?

6.       Line 379 – “between Japan and Western” – please use a different word instead of “western”

7.       Line 413 – “As a contributing factor in determining the immune status, the expression of PD-L1 in tumor cells, which is involved in the cancer immune evasion mechanism, may play a role.”

To me this is the main point of the conclusion, but I think that it has to be changed. It is not enough to sum up the entire study with a statement like this: “PD-L1 (…) may play a role (…)”. Please rephrase this statement so that it is more specific.

only some very minor changes required

Author Response

Please see the attachment." in the box 

Reviewer 2 Report

This is an interesting study describing how the immunological components of the tumor microenvironment and markers of inflamation influence thyroid tumor progression via PD-L1 protein. Although the number of samples is very limited, the study was well designed and the conclusions seem sound. However, some parts of the manuscript require clarifications in order to be more comprehensible for the readers.

Major concerns:

1. On Figure 1d, I can see cells expressing PD-L1 (stained red). How is their percentage 0%?

2. On Figure 2, probably due to image quality, one can not differentiate between orange and purple circles.

3. The legend of Fig2 seems to be wrong. Is it depisting the staining of CD8 or PD-L1?

Minor concerns:

1. In the introduction section (page 2, line 55), it should be explained that pembrolizumab is an antibody against PD-L1. I presume that most readers are not familiar with this fact.

2. The role of CD8 expression on T-cell behaviour should be explained.

3. In the Material and Methods section, it should be stated that both blood and archival samples were collected from all 33 patients (were they?)

4. In the Material and Methods section, section 2.3., lines from 134-138 are confusing. The comparison of stainability is much better explained in the Discussion section (page 16, line 330). I suggest rewriting. Also, Tumor proportion score (TPS) is not defined.

Needs moderate improvements

Author Response

Please see the attachment." in the box 

Reviewer 3 Report

Sekino et al. investigated the expression of PD-L1, the presence of the BRAF V600E mutation, and immune factors in thyroid cancer, as well as their association with prognosis.

They reported that the presence of BRAFV600E mutation and positive expression of CD8 were associated with PD-L1 positivity. Although no clear associations were showed, there was a tendency for better prognosis when there was a higher infiltration of CD8+ T cells within the tumor.

Major comments:

- More conservative and personalized treatment options have been developed in recent years in thyroid cancer. This point together with the need of detecting new prognostic factors should be briefly described in the introduction. Please, see and cite: PMID: 33213119

- The aims, design and conclusions of the study is not very clear. Authors reported “The aim of this study is to characterize the expression of PD-L1 and the presence of …., and to investigate the factors that predict the effectiveness of treatment with anti-PD-L1 antibody therapy.” Results related to this last aim are not reported and discussed. Authors should report the survival of patients with thyroid cancer after treatment with PD-L1 antibodies. Please, specify the types of treatments (immunotherapy, etc) received by the population enrolled in this study and modify conclusions accordingly.

- The authors should better underline the main novelty of the present manuscript compared to previous studies.

- I would add a short paragraph on future perspectives. This section should include:

1)      The future development of predictive models combining clinical, histopathological and/or genetic data to define the risk class of patients.

2)      The potential role of liquid biopsy in this context (please, see and cite: PMID: 33213118).

Minor comments:

- Please, check the use of acronyms. An abbreviation should be used only if the term appears at least 4-5 times in the main text. If the term is used only few times it should not be abbreviated.

- The spelled-out form of each acronym should be indicated on first use.

Author Response

Please see the attachment." in the box 

Round 2

Reviewer 1 Report

All my commentaries have been satisfactorily replied to.

The only point that I find hard to accept is the explanation about the cut-off value for the age of the patients. I think that it is important for two reasons:

-    The cut-off value of 55 years should be applied to match the latest guidelines (nobody uses the cut-off value used by authors)

-    The second, most important reason is that the change of this cut-off value can influence the results of the study, and thus change the conclusions of the work

This is the commentary I’m referring to:

1. Table 2, patients characteristics:
Age <50 vs ≥50 years – why do authors use this cut-off value for patient’s age?
Tumor size <20 vs ≥20 (mm) – why this cut-off value for tumor size?

Response:
We thank the reviewer for this pertinent comment.

The current UICC-TNM classification and AJCC TNM classification, introduced in 2016 and 2017 respectively, raised the cutoff value for age from 45 to 55 years. Consequently, in 2019, Japan also revised its guidelines for the management of thyroid cancer and raised the age boundary in the TNM classification (8th edition) to 55 years from the previous 45 years. The data was collected shortly before the age classification was revised in Japan. Considering the trends in thyroid cancer classification, the cutoff value for age was set at 50 years.

Author Response

Thank you for your suggestion. We agree with your comment and changed the cut-off value to 55 years old as shown in Table 2. (the part of yellow marker in Table 2)

Even when the cutoff value was changed to 55 years old, there was no significant association with PFS (Progression-Free Survival) in the univariate analysis. Therefore, results and conclusions have not be revised.

Reviewer 3 Report

The paper has reached a high enough priority to be acceptable for publication 

Author Response

We would like to thank the reviewers for your insightful comments, which have greatly helped us to improve the quality of our manuscript.

Round 3

Reviewer 1 Report

"All my commentaries have been satisfactorily replied to.
I now consider the study eligible for publication"